# Integrating Citizens' Importance-Performance Aspects into Sustainable Plastic Waste Management in Danang, Vietnam

Thi Thanh Thuy Phan [1,2], Van Viet Nguyen [1,3], Hong Thi Thu Nguyen [4] and Chun-Hung Lee [1,*]

1   Department of Natural Resources and Environmental Studies, College of Environmental Studies and Oceanography, National Dong Hwa University, Hualien 97401, Taiwan
2   Faculty of Natural Resources & Environment, Dong Nai Campus, Vietnam National University of Forestry, Bien Hoa 810000, Vietnam
3   Faculty of Silviculture, Dong Nai Campus, Vietnam National University of Forestry, Bien Hoa 810000, Vietnam
4   Faculty of Chemistry, The University of Danang—University of Science and Education, 41 Le Duan Road, Danang 550000, Vietnam
*   Correspondence: chlee@gms.ndhu.edu.tw; Tel.: +886-3-8903343

**Abstract:** Plastic pollution is a matter of deep concern that requires an urgent and international response, involving stakeholders at all levels. The rapid increase of single-use plastic and medical waste, especially in the context of COVID-19, has caused a drastic progression in the plastic pollution crisis on a global scale. To identify an efficient plastic waste management (PWM) system to tackle this major environmental problem, this study adopted importance-performance analysis and used logistic regression to identify key factors affecting citizens' behavior to participate in PWM strategies in Vietnam. The results indicate that while the importance of all PWM solutions was considered to be high, their performance was rated at a low level, implying a sizable gap between perceived importance and performance of eleven solutions for PWM. The findings also show that solutions such as "offering zero-waste lifestyle seminars to citizens", "having community engagement", "using eco-friendly products", and "imposing a ban on single-use plastics" are useful for the development of an effective environmental policy. Furthermore, it was found that the following characteristics have a significant influence on citizens' participation in PWM solutions: (1) gender, (2) education level, (3) residential area, (4) employment status, and (5) citizens' awareness and behavior towards plastic reduction. This study is expected to provide theoretical and empirical evidence for policymakers and authorities who are in charge of promulgating the necessary mechanisms and policies to promote the socialization of PWM.

**Keywords:** community engagement; citizens' perceptions; importance-performance analysis; institutional incentives; plastic waste management; willingness to participate

## 1. Introduction

During recent decades, plastic pollution has become an urgent environmental issue and has gained considerable attention worldwide [1,2]. It is estimated that 300 million metric tons (MMT) of plastic waste (PW) is created every year [3]. In particular, the recent COVID-19 pandemic has resulted in an intensified demand for single-use plastics (SUPs), adding to the already uncontrollable global PW crisis [4]. In 2021, more than 8 million tons of pandemic-related PW was generated worldwide, with approximately 25,000 tons going into the world's oceans [5]. Globally, approximately 16% of PW is recycled, 25% is incinerated, and more than 40% of PW is disposed in landfills, dumps, or directly in the environment [3]. Plastic pollution in seas and coastal areas is a serious problem, as plastic consumption and disposal at current rates will have devastating repercussions for marine life and ocean health [6].

An intractable problem with the current level of PW is that appropriate sites for landfilling are exhausted, and treatment infrastructure and technology are still lacking in developing countries [7,8]. Take Vietnam as an example: this country is not only experiencing increasing urbanization and population growth, but as one of the world's major importers of plastic scrap, the country also faces waste management problems, particularly when dealing with PW [9]. In Vietnam, about 0.35–0.78 million tons of PW are discharged into the environment each year, accounting for 16.0–23.0% of the total waste in landfills [10,11]. Similar to developed countries, Vietnam is currently aiming to improve the quality of plastic waste management (PWM) services to mitigate uncontrolled or illegal disposal. However, lack of waste management infrastructure, reliance on low-tech machinery, and limited administration and financial resources from municipal authorities have all led to inadequacies in the management of PW [12–14]. Unsurprisingly, inadequate waste management systems have affected social life, marine ecosystems, biodiversity, and the environment [6,8,15].

It is clear that effective PWM is important for not only human health, wildlife, ecosystems, and the marine environment, but also for sustaining environmental, social, and economic well-being. Many researchers have described a range of strategies and applications for minimizing PW [16], such as the 3Rs (reduce, reuse, and recycle), waste collection infrastructure improvement [7], the implementation of a ban on plastic bags [17–19], and tax policies [20,21]. Furthermore, other recommendations, such as improving citizens' awareness [22], using eco-friendly products [23,24], and incorporating community engagement [25,26], have also been widely discussed in the literature. However, there are still fundamental challenges in terms of the sustainability of these approaches in terms of addressing the inconsistencies of PWM [27–29]. More specifically, PWM represents a complex strategic issue that is limited by space for disposal, time requirements, and lack of consistency in expected outputs [30,31]. Thus, in order to carry out PWM successfully, it is critical to understand how to better assess the relevant aspects and criteria. As a result, determining the importance and performance (I-P) of solutions for PWM is a topic of pressing concern for researchers [32].

Importance-performance analysis (IPA) is one of the most popular methodological tools in research fields including tourism, business, and management [33,34]. IPA is not only used to test the importance of a given item, but also to determine the performance of the item or factor in the citizens' level of satisfaction or their subjective evaluation of the item [32,35]. This means IPA is an effective assessment technique for identifying positive attributes, as well as those needing to be improved upon, and for which quick action is required [36]. It is also used to determine discrepancies between what stakeholders think is an important component of a specific problem, and their actual perceptions of how well the problem is being managed [33,37]. Many studies on waste management and tourism have been conducted using the IPA model, such as investigations by Bui et al. [35], Boley et al. [34], Lai and Hitchcock [33], and Tseng [32]. However, to the best of our knowledge, no research has been performed to date on devising a conceptual framework for PWM in Danang, Vietnam, taking into account citizens' perspectives as revealed by IPA. In addition, the differences in urban and rural citizens' perceptions of the I-P levels of various PWM solutions has not been considered. By using the IPA model initially developed by Martilla and James [37], this paper aims to determine the relative importance of the attributes related to PWM solutions and the degree of citizens' satisfaction with the performance of these attributes. Moreover, the results of an IPA of PWM solutions can serve as a strategic decision-making tool for those charged with implementing solutions to improve the effectiveness of PWM systems. The contribution of this study is unique, as it investigates citizens' viewpoints on PWM solutions to provide insights for future directions for PWM and its policy. In order to fulfill the above-mentioned rationale underlying the research, the main objectives of this study are threefold. Firstly, this study establishes distinct sets of indicators for citizens' plastic management solutions, based on a literature review, and collective viewpoints obtained from citizens' replies in Danang, Vietnam.

Second, it measures the I-P levels of the respective perception indicators among urban and rural citizens. Finally, the study analyzes the factors that influence citizens' participation via the development of logistic regression models (LRM) and builds an evaluation framework for PWM strategies. All types of PW, including polyethylene and biodegradable plastic components, were included in the study. Biodegradable plastic components are those made from protein, silk, wool, or plant cellulose or starch.

## 2. Literature Review

### 2.1. Integrating the Perspectives of Community Engagement and Institutional Incentives in Order to Work toward PWM Solutions

Previous studies have mainly focused on addressing single solutions such as recycling, raising community awareness, developing infrastructure, or implementing policies [7,16,19,38]. Willis et al. [2] have stated that a sustainable PWM system needs to integrate solutions that range from time-tested and traditional, to cutting-edge and innovative. Education and training on the segregation and recycling of PW (code I1) gives citizens knowledge, skills, and better awareness of how to dispose of used plastic products safely and properly [22]. Previous studies have also revealed that offering zero-waste lifestyle seminars to citizens (code I2) is essential for PWM practices, including plans to keep PW out of landfill sites, incinerators, and oceans, and to protect the environment for present and future generations [39]. In addition, infrastructure accessibility plays an important role in waste collection and transportation, and developing it has a favorable impact on system efficiency [7]. Rai et al. [7] have argued that the availability of equipment for waste separation is a prerequisite for citizens' willingness to classify waste. It has also been shown that providing segregation and recycling bins (code I3) is necessary for the coordination of waste management between households and the local government [40]. Additionally, establishing collection and recycling stations (code I4) promote household waste separation and recycling, as well as collaboration among stakeholders for better waste control and management [7].

Sustainability can be a major challenge as communities continue to generate more waste due to population growth and current needs [41]. Environmental protection procedures may lead to sustainable utilization. For example, using cloth bags and reusable containers (code I5) has been shown to mitigate pollution and the volume of waste plastic bags in the waste management chain [24,42]. Gutt and Amariei [27] have stated that using eco-friendly and biodegradable products (code I6) has a beneficial effect on the natural environment, ecosystems, oceans, and wildlife. Gill et al. [23] have indicated that the use of eco-friendly products was being promoting more frequently as a reflection of businesses taking responsible end-of-life product management into consideration.

Community engagement is not only an important component of PWM development strategies, but also a foundational element of modern social life. Community engagement can reduce inappropriate waste disposal behavior, and, in turn, contribute to environmental improvement [25,43]. Indeed, fostering community engagement has emerged as a common approach to implementing waste management strategies in developing countries. In particular, collaboration among citizen associations (code I7) has proven to be economically advantageous due to the separation and recycling of waste at the source by the community [43–45]. Such activities and involvement could enhance the citizenry's level of environmental awareness and influence their proactive behaviors towards sustainable PWM. Collaboration between non-governmental organizations (NGOs) and government agencies (code I8) is a widely employed approach to providing waste management services [25,26,46,47]. In addition, effective policy tools are necessary for PWM strategies. Thus, the role of legislation and regulations in the development of any PWM system cannot be ignored. In terms of policy, imposing fees for plastic bags at supermarkets (code I9) is one approach to encouraging individuals to change their behavior, born of a rationale that curbing the consumption of plastic bags will reduce environmental pollution [20,21,48]. Nguyen et al. [49] and Zorpas et al. [50] have pointed out that schools and state agencies implementing the SUPs ban (code I10) could facilitate a migration from SUPs to reusable

alternatives, and enhance citizens' awareness in the process. Policies banning SUPs at supermarkets and restaurants (code I11) have been demonstrated to minimize the use of plastic, which benefits the environment [19]. Building on these findings in previous literature, the present research suggests a set of attributes that includes five aspects: "education and information", "infrastructure accessibility", "environmentally friendly alternatives", "community engagement" and "legislative tools", which are adapted into 11 indicators (listed in Table 1).

**Table 1.** Indicators of PWM solutions.

| Aspects | Code | Indicators (Abbreviations) | References |
|---|---|---|---|
| Education and information | I1 | Training on segregation and recycling of plastic waste (segregation and recycling) | [22,51] |
| | I2 | Offering zero-waste lifestyle seminars to citizens (zero-waste) | [39,52] |
| Infrastructure accessibility | I3 | Providing segregation and recycling bins (providing bins) | [40] |
| | I4 | Establishing collection and recycling stations (establishing stations) | [7] |
| Environmentally friendly alternatives | I5 | Using cloth bags and reusable containers (cloth bags and containers) | [24,42] |
| | I6 | Using eco-friendly and biodegradable products (eco-friendly products) | [23,27] |
| Community engagement | I7 | Collaboration among citizen associations (citizen associations) | [43–45] |
| | I8 | Collaborations between NGOs, government agencies (NGOs, government agencies) | [25,26,46,47] |
| Legislative tools | I9 | Imposing fees for plastic bag use at supermarkets (fees) | [21,48] |
| | I10 | Implementing SUPs ban in schools and state agencies (implement SUPs ban) | [49,50] |
| | I11 | Imposing a ban on SUPs at supermarkets and restaurants (imposing SUPs ban) | [17–19] |

### 2.2. Factors Affecting Citizens' Participation Behavior in PWM Solutions

Several recent studies have determined factors influencing citizens' awareness and behavior toward PWM. Research by Wang et al. [53] and Wang et al. [54] points out that waste segregation is essential for solving the current dilemma of waste management. Citizens' participation in proper waste segregation could mitigate one-third of all waste, and certain components of waste could be reused and remanufactured to create new products [22,51,55]. In other studies, Matiiuk and Liobikienė [56] and Zhang et al. [55] assessed citizens' behavior in terms of their participation in environmental protection activities. They suggest that individual-level attributes (such as attitudes and awareness), organizational factors (such as management support and resources), and external factors (such as economic conditions and government policy) play pivotal roles in the success of PWM strategies. Also, there is evidence that shows that the demographic characteristic of citizens and their level of environmental awareness affect the participation of individuals in PWM solutions [57]. Moreover, citizens' awareness of PW, which has been found to vary according to their gender, education level, employment, and residential area (i.e., urban versus rural citizens), appears to be a key factor influencing individuals' behavior with regard to their participation in PWM solutions [22,53,58]. Work by Chung and Yeung [59] points out that the government should focus on measures appealing to morality (i.e., promoting waste reduction as a component of sustainable waste management and health protection) rather than focusing on economic means, to motivate the group of people they studied to reduce waste. Similarly, Ekere et al. [60] have indicated that individuals' membership of environmental organizations was linked to a higher willingness to participate (WTP) in waste

management strategies. In general, understanding citizens' demographic characteristics and participation behaviors is crucial for developing PWM strategies, as this knowledge helps to guarantee that programs will be sustainable. Of course, this is best achieved when citizens' needs and concerns are heard and integrated into these efforts.

## 3. Methodology

### 3.1. Study Area

Danang, the selected study location, is one of Vietnam's five main cities, and is located on the South China Sea coast (Figure 1). This area covers eight administrative districts, with a total area of 1285.4 km$^2$ and a population of 1,134,310 in 2020 [61]. According to the most recent census, 87.2% of Danang's population lives in urban areas, with an annual urban population growth rate of 3.5% [61]. The expansion of Danang's population and its burgeoning economy have increased the quantity of waste generated in the city, putting pressure on the city's waste management capacity as it struggles to deal with 1100 tons of solid waste each day [8]. However, it is estimated that only 8–10 % of total waste is collected by the informal system of independent waste pickers, collectors, or unregistered units [8]. Plastic pollution has emerged as a major environmental concern in Danang in recent years, mirroring prioritization of the issue at the national level. In particular, Danang is hampered by inadequate waste treatment, owing to a lack of waste treatment technology and low capacity of waste collection and transportation vehicles [8,35,62], all of which impedes PWM. Hence, the city's leaders urgently need to adopt PWM strategies to bolster their decision-making framework. Therefore, there is a compelling need for the present study and to develop an evaluation framework for PWM in Danang using IPA.

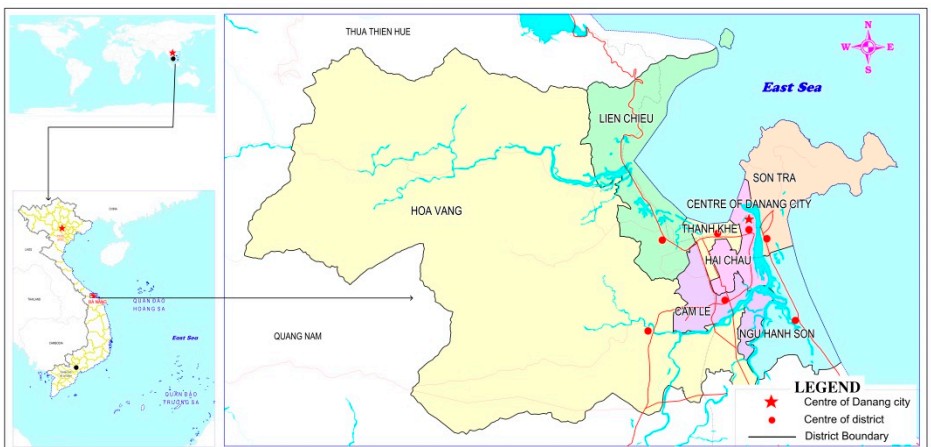

**Figure 1.** Location map of the study area in Danang, Vietnam.

### 3.2. Questionnaire Design and Survey Execution

The present study is constructed as an explorative study with a focus on citizens' perspectives towards PWM solutions for the protection of the environment. Hence, a questionnaire was developed based on the literature review, group interviews with stakeholders (including scholars, educators, policymakers, and NGOs), and findings gleaned from a pre-test survey about PWM solutions. A total of 11 indicators pertaining to PWM solutions were included in the questionnaire (Table 1). The formal questionnaire begins with an introduction to the purpose of the survey and description of the research, which is broken down into three main sections. Respondents were provided with a privacy statement to ensure the anonymity of their responses. The participants could also get support from the researcher if they had any questions or needed clarification on any questions. The survey was conducted in the Vietnamese language, and written in simple, understandable terms. The first section consists of issues related to respondents' perceptions and attitudes towards PW (i.e., separation of waste at home, participation in environmental protection activities, PW disposal sites, using eco-friendly products, WTP in PWM programs). The

second section focuses on evaluating citizens' perceptions of the I-P of PWM solutions. In this section, respondents were asked to give their perceptions of the I-P of 11 indicators on a 5-point Likert scale ranging from 5 (very important/strongly satisfied) to 1 (very unimportant/strongly dissatisfied) [63,64]. The final section gathers respondents' socio-demographic information (i.e., age, gender, household size, education level, income, employment, and residential area).

Face-to-face interviews with citizens were conducted in June 2021 in 7 selected districts. 572 citizens were selected by systematic random sampling for the interviews. Complete responses were collected from 561/572 citizens, achieving a response rate of 98.07%, with only 1.93% of responses excluded since they were not fully completed or contained inconsistent answers. At the survey sites, for each selected household, a citizen over 20 years old was interviewed. It should be noted that the sample size in each district was decided based on the population characteristics of those districts.

*3.3. Data Analysis*

The collected data were analyzed using SPSS software (version 26; SPSS Statistics Inc., Chicago, IL, USA). First, citizens' socio-demographic data and their awareness and behaviors around PWM were analyzed. This was followed by the calculation of the matrix framework to assess the mean scores and ranking of I-P indicators of PWM solutions. In the next stage of analysis, the statistically significant differences between respondents' perceptions of the I-P of PWM solutions were evaluated using paired sample *t*-tests where $p < 0.05$ indicated significance.

After that, IPA was applied in the study, as described by Martilla and James [37], to map the data to the IPA grid. In this grid, the I-P mean scores for PWM solutions are plotted on the vertical and horizontal axes, as depicted in Figure 2. The X-axis illustrates citizens' perceptions of the performance, while the Y-axis illustrates their view of an attribute's importance with respect to the indicators of PWM solutions. Indicators in quadrant I (Keep up the good work) are rated as "very important" by respondents, so responsible organizations must ensure they maintain their current level of performance, at a minimum. Indicators in quadrant II (Concentrate Here) are perceived by the respondents as "very important but performing poorly". This indicates that improvement efforts should be concentrated on indicators located here. Indicators in quadrant III (Low Priority) are perceived by respondents as "less important" and their performances level is likewise fairly low. Quadrant IV (Possible Overkill) contains indicators that are not perceived as important, but their performance is nonetheless assessed to be good by respondents. As a result, the location of each indicator clearly shows the properties of specific PWM solutions as perceived by citizens.

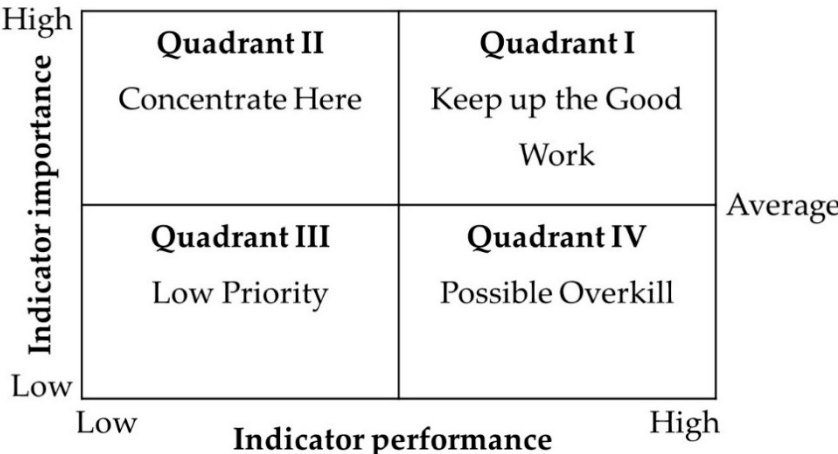

**Figure 2.** Evaluation grid for IPA (modified from Lai and Hitchcock [33]).

Finally, based on binary choice theory, an LRM was applied to investigate citizens' participation in, perception of, and behavior towards PWM solutions. Logit and probit models

were applied to explore how demographic variables (i.e., gender, education, employment, and residential area), perceptions of and behaviors toward PWM, and perceived I-P of PWM solutions affected citizens' WTP in the PWM program. Four models were developed. The first model (Model I) is a logit model, which takes the interviewers' "WTP in PWM program" as the dependent variable, and considers the demographics variables, awareness of and behaviors towards PW, and the overall importance means of PWM solutions as the independent variables. The second model (Model II) is also a logit model with all variables kept similar to those in the first model, but with the overall performance mean of the PWM solutions included instead of the overall importance mean. Model III and Model IV are similar to Model I and Model II, respectively, but are probit models.

## 4. Results

### *4.1. Respondents' Social Backgrounds and Environmental Awareness and Behaviors*

The descriptive analysis of participants' demographic variables and their environmental awareness and behaviors is shown in Table 2. Among 561 respondents, the proportion of females (59.5%) was higher than that of males (40.5%). More than half of respondents (59.9%) had an average age of 30–49 years, and a majority of citizens (70.8%) were educated at the college/vocational level or higher. A significant proportion of respondents (61.7%) reported a personal income between 4,000,000–10,000,000 Vietnamese Dong (VND) per month, while only a very small number of respondents (11.6%) reported an income of 10,000,000 VND or more per month. Out of 561 respondents, only 33.2% implemented waste separation at home, while over 90.7% of the respondents participated in environmental protection activities. The majority of respondents prioritized the use of eco-friendly materials (62.4%) and expressed their WTP in a PWM program (71.8%).

**Table 2.** Demographic characteristics, environmental awareness and behaviors of respondents.

| Characteristics | All Citizens | | Urban Citizens | | Rural Citizens | |
|---|---|---|---|---|---|---|
| | Frequency | % | Frequency | % | Frequency | % |
| Number | 561 | | 444 | | 117 | |
| Male | 227 | 40.5 | 170 | 38.3 | 57 | 48.7 |
| Female | 334 | 59.5 | 274 | 61.7 | 60 | 51.3 |
| Single | 172 | 30.7 | 135 | 30.4 | 37 | 31.6 |
| Married | 389 | 69.3 | 309 | 69.6 | 80 | 68.4 |
| Age | | | | | | |
| 20–29 | 89 | 15.9 | 69 | 15.5 | 20 | 17.1 |
| 30–39 | 194 | 34.6 | 158 | 35.6 | 36 | 30.8 |
| 40–49 | 142 | 25.3 | 108 | 24.3 | 34 | 29.1 |
| 50–59 | 73 | 13.0 | 61 | 13.7 | 12 | 10.3 |
| ≥60 | 63 | 11.2 | 48 | 10.8 | 15 | 12.8 |
| Education | | | | | | |
| Lower secondary | 37 | 6.6 | 28 | 6.3 | 9 | 7.7 |
| Upper secondary | 127 | 22.6 | 93 | 20.9 | 34 | 29.1 |
| College/Vocational education | 102 | 18.2 | 79 | 17.8 | 23 | 19.7 |
| University | 254 | 45.3 | 212 | 47.7 | 42 | 35.9 |
| Master's degree or above | 41 | 7.3 | 32 | 7.2 | 9 | 7.7 |
| Monthly income (VND/household/month) | | | | | | |
| Less than 4,000,000 | 150 | 26.7 | 120 | 27.0 | 30 | 25.6 |
| 4,000,000–7,000,000 | 238 | 42.4 | 187 | 42.1 | 51 | 43.6 |
| 7,000,000–10,000,000 | 108 | 19.3 | 80 | 18.0 | 28 | 23.9 |
| 10,000,000–13,000,000 | 50 | 8.9 | 45 | 10.1 | 5 | 4.3 |
| Above 13,000,000 | 15 | 2.7 | 12 | 2.7 | 3 | 2.6 |
| Environmental awareness and behaviors | | | | | | |
| Separation of waste at home (Yes) | 186 | 33.2 | 154 | 34.7 | 32 | 27.4 |
| Participating in environmental protection activities (Yes) | 509 | 90.7 | 404 | 91.0 | 105 | 89.7 |
| Using plastic alternatives materials or eco-friendly (Yes) | 350 | 62.4 | 274 | 61.7 | 76 | 65.0 |
| Willingness to participate in PWM program (Yes) | 403 | 71.8 | 340 | 76.6 | 63 | 53.8 |

VND: Vietnam Dong (1 USD = 23,077 VND) (data from The State Bank of Vietnam on 22 June 2021).

*4.2. Matrices of I-P Levels of PWM Solutions*

Table 3 is a matrix that illustrates the mean scores, the top, and the bottom three ranks of I-P of PWM solutions for urban and rural citizens, respectively. The results of Table 3 show that all respondents collectively, as well urban and rural citizen cohorts individually, assigned a high level of importance to all PWM solutions, with overall mean scores of 4.253, 4.119, and 4.225, respectively. However, all respondents expressed their belief that the indicators related to PWM solutions performed poorly, with the overall performance means for urban citizens, rural citizens, and all respondents being 2.314, 2.350, and 2.322, respectively. The importance levels of the indicators were significantly greater than their performance levels, with a difference of over 1.051 points (the I-P difference is shown in Table 3).

**Table 3.** Mean scores and paired-sample *t*-test of I-P levels of citizens.

| Code | Indicator | Importance Mean (Rank) | Performance Mean (Rank) | Difference(I-P) | T-Value | Sig. (2-Tailed) |
|------|-----------|------------------------|-------------------------|-----------------|---------|-----------------|
| All respondents (n = 561) | | | | | | |
| I1 | Segregation and recycling | 4.257 | 3.162 (1) | 1.094 | 61.970 | 0.000 |
| I2 | Zero-waste | 4.282 (1) | 2.232 | 2.050 | 85.027 | 0.000 |
| I3 | Providing bins | 4.248 | 3.023 (2) | 1.225 | 51.490 | 0.000 |
| I4 | Establishing stations | 4.275 (2) | 2.105 | 2.169 | 92.468 | 0.000 |
| I5 | Cloth bags and containers | 4.253 | 2.809 (3) | 1.444 | 61.981 | 0.000 |
| I6 | Eco-friendly products | 4.180 (10) | 2.086 | 2.094 | 88.492 | 0.000 |
| I7 | Citizen associations | 4.232 | 2.283 | 1.948 | 111.745 | 0.000 |
| I8 | NGOs, government agencies | 4.269 (3) | 2.146 | 2.123 | 108.394 | 0.000 |
| I9 | Fees | 4.007 (11) | 2.000 (9) | 2.007 | 66.996 | 0.000 |
| I10 | Implement SUPs ban | 4.242 | 1.982 (10) | 2.260 | 88.266 | 0.000 |
| I11 | Imposing SUPs ban | 4.228(9) | 1.711 (11) | 2.517 | 105.713 | 0.000 |
| | Overall mean | 4.225 | 2.322 | | | |
| Urban citizens (n = 444) | | | | | | |
| I1 | Segregation and recycling | 4.275 | 3.169 (1) | 1.106 | 61.294 | 0.000 |
| I2 | Zero-waste | 4.315 (1) | 2.218 | 2.097 | 81.301 | 0.000 |
| I3 | Providing bins | 4.261 | 3.038 (2) | 1.223 | 51.969 | 0.000 |
| I4 | Establishing stations | 4.291 (3) | 2.097 | 2.194 | 86.900 | 0.000 |
| I5 | Cloth bags and containers | 4.273 | 2.786 (3) | 1.486 | 58.985 | 0.000 |
| I6 | Eco-friendly products | 4.255 (9) | 2.081 | 2.173 | 87.395 | 0.000 |
| I7 | Citizen associations | 4.257 | 2.297 | 1.959 | 112.878 | 0.000 |
| I8 | NGOs, government agencies | 4.306 (2) | 2.137 | 2.169 | 106.012 | 0.000 |
| I9 | Fees | 4.032 (11) | 1.977 (9) | 2.054 | 65.967 | 0.000 |
| I10 | Implement sups ban | 4.261 | 1.957 (10) | 2.304 | 86.376 | 0.000 |
| I11 | Imposing sups ban | 4.255 (9) | 1.700 (11) | 2.554 | 94.491 | 0.000 |
| | Overall mean | 4.253 | 2.314 | | | |
| Rural citizens (n = 117) | | | | | | |
| I1 | Segregation and recycling | 4.188 (3) | 3.137 (1) | 1.051 | 21.100 | 0.000 |
| I2 | Zero-waste | 4.154 | 2.282 | 1.872 | 31.782 | 0.000 |
| I3 | Providing bins | 4.197 (2) | 2.966 (2) | 1.231 | 17.290 | 0.000 |
| I4 | Establishing stations | 4.214 (1) | 2.137 | 2.077 | 35.556 | 0.000 |
| I5 | Cloth bags and containers | 4.179 | 2.897 (3) | 1.282 | 23.131 | 0.000 |
| I6 | Eco-friendly products | 3.897 (11) | 2.103 | 1.795 | 32.625 | 0.000 |
| I7 | Citizen associations | 4.137 | 2.231 | 1.906 | 37.025 | 0.000 |
| I8 | NGOs, government agencies | 4.128 (8) | 2.179 | 1.949 | 39.112 | 0.000 |
| I9 | Fees | 3.915 (10) | 2.085 (9) | 1.829 | 22.900 | 0.000 |
| I10 | Implement SUPs ban | 4.171 | 2.077 (10) | 2.094 | 30.990 | 0.000 |
| I11 | Imposing SUPs ban | 4.128 (8) | 1.752 (11) | 2.376 | 49.357 | 0.000 |
| | Overall mean | 4.119 | 2.350 | | | |

Urban citizens rated the indicators "zero-waste" (code I2), "NGOs, government agencies" (code 8), and "establishing stations" (code I4) as the most important indicators. They

gave the highest performance ranking to the indicator "segregation and recycling" (code I1), followed by "providing bins" (code I3), which was followed by "cloth bags and containers" (code I5). In contrast, they assigned the lowest performance ranking to the indicator "imposing SUPs ban" (code I11). The indicators "fees" and "implement SUPs ban" were both considered by urban citizens as being relatively unimportant, and performing poorly (code I9, I10). Notably, urban citizens' ratings of the I-P PWM solutions were consistent with those of all respondents (see Table 3).

Rural citizens, given the same set of indicators, perceived the indicators of "providing bins", "segregation and recycling", and "cloth bags and containers" as having the highest importance and performance ratings of all indicators. In contrast, they rated the I-P of "eco-friendly products", "fees", "NGOs, government agencies" and "imposing SUPs ban" as lower, relative to the I-P of the other indicators. Noticeably, rural citizens perceived "establishing stations" and "implement SUPs ban" as more important than other indicators, but they rated these methods to be lower in terms of performance.

Based on paired-sample *t*-tests, it was found that all indicators exhibited significant differences between importance and performance levels among the urban citizens, rural citizens, and all respondents' cohorts (all indicators) (Table 3). This implies that significant room exists for improvement.

### 4.3. IPA of Citizens' Perceptions of PWM Solutions

A visual analysis of the IPA rating grid plots across the four quadrants for the proposed PWM solutions, shown in Figure 3, shows the relative strengths and weaknesses of the indicators of I-P. Based on the views expressed by urban citizens, indicators I1, I3, and I5 were located in quadrant I. These findings imply that environmental managers and policymakers should continue to maintain their current performance in order to meet the goal of advancing sustainable PWM in the near future. On the other hand, there were seven indicators assigned to quadrant II based on urban citizens' ratings; namely, "zero-waste", "establishing stations", "eco-friendly products", "citizen associations", "NGOs, government agencies", "implement SUPs ban", and "imposing SUPs ban". Evidently, most urban citizens believe that there is room for improvement in the performance of these PWM solutions, and that a combination of these solutions should be implemented in the future. This points to the fact that providing the PWM performance for a sustainable development strategy must be considered a top priority. Only the indicator "fees" (code I9) was located in quadrant III (Low Priority) by this cohort, which means urban citizens perceived this indicator to be relatively unimportant and also felt this solution was not performing well. Similarly, the I-P analysis of rural citizens also showed that two indicators were located in quadrant I, one in quadrant II, seven in quadrant III, and one in quadrant IV (Figure 3). Various indicators were determined to fall within quadrant I, indicating that these solutions are perceived as very important and showing a high level of satisfaction with their performance. Therefore, managers should "keep up the good work" with regard to this indicator, continuing programs such as "segregation and recycling" (code I1), and "providing bins" (code I3). The indicator "establishing stations", located in quadrant II, should be "concentrated" upon for improvement because it has high importance but low performance in the eyes of rural citizens. Martilla and James [37] called the third quadrant "low priority"; accordingly, the indicators in this zone were considered relatively less important and their performances were lower than the mean score of other indicators. There are seven indicators in this quadrant in the present study, including "zero-waste" (code I2), "eco-friendly products" (code I6), "citizen associations" (code I7), "NGOs, government agencies" (code I8), "fees" (code I9), "implement SUPs ban" (code I10), and "imposing SUPs ban" (code I11). Finally, "cloth bags and containers" (code I5) was located in quadrant IV. The importance of this solution was deemed to be low, but it was considered to perform well, which could indicate overinvestment and a misplaced overabundance of concern with this measure.

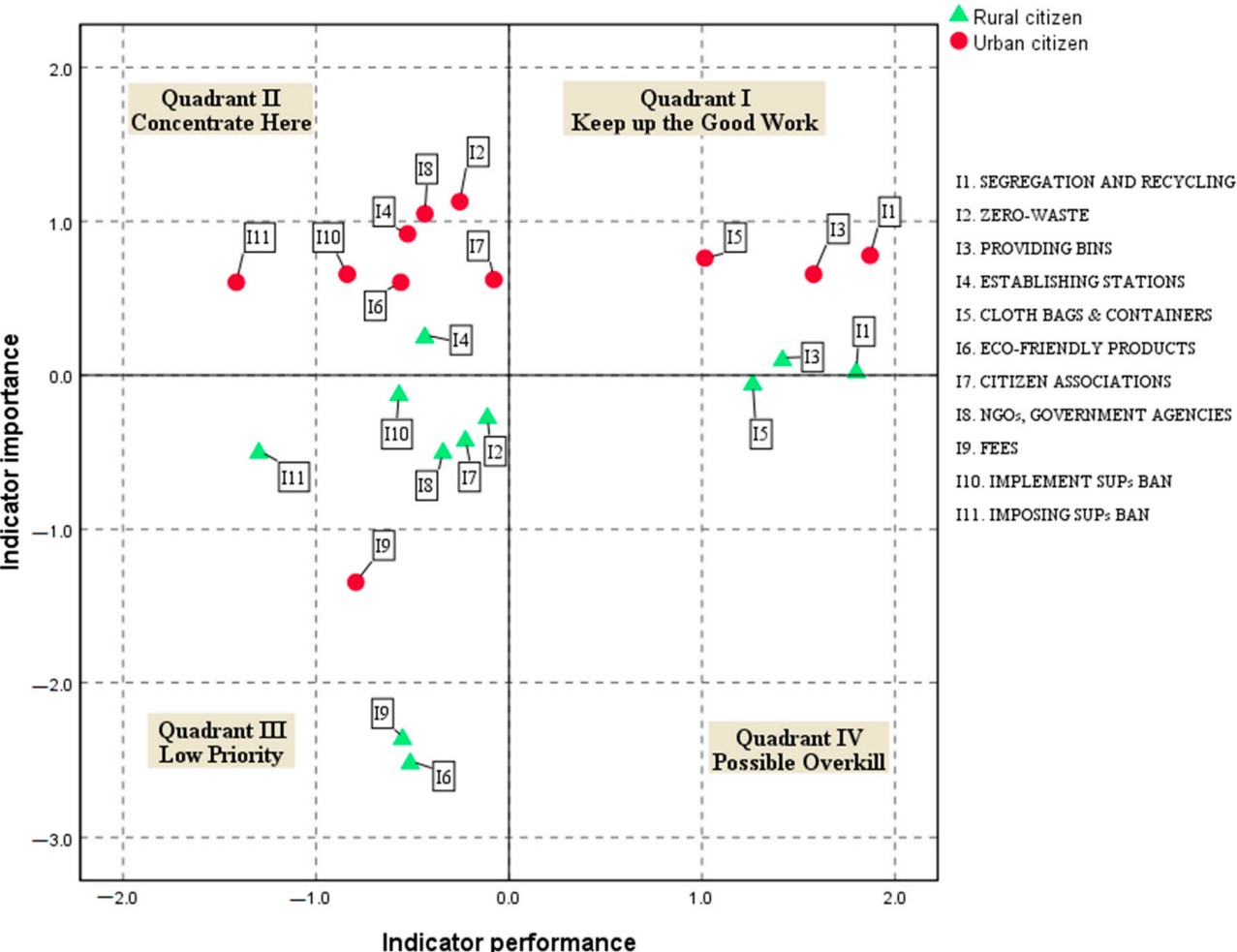

**Figure 3.** Importance-performance plot of PWM solutions between urban and rural citizens.

*4.4. Citizens' Participation Behavior in PWM Solutions*

Information about citizens' participation behavior in PWM solutions can help develop and improve waste management systems. To evaluate citizens' behavior in the context of PWM, we used citizens' WTP in PWM solutions as the dependent variable in this study, and considered citizen's socio-economic backgrounds (i.e., gender, education, employment, and residential area), awareness and behavior towards PWM (i.e., waste separation, participating in environmental protection activities, using eco-friendly products, and being a member of environmental organizations or clubs), and the overall I-P means of PWM solutions as the independent variables. In this study, the above dependent and independent variables were integrated into logit and probit regression models for comparison, with the results presented in Table 4. Those results show that the −2Log likelihood and Akaike information criterion (AIC) produced robust results with acceptable levels [65]. With regard to the Hosmer and Lemeshow test of the goodness of fit, both logit and probit models are appropriate (much greater than 0.05). Moving on, the Chi-square ($\chi^2$) is highly statistically significant, with a value of 21.66 and df = 9, indicating that our modeling of citizens' participation behavior in PWM solutions gives solid results within the model specifications.

**Table 4.** Estimation results of citizens' perception of PWM participation behavior.

| Variable Names | Logit Model | | | | Probit Model | | | |
| --- | --- | --- | --- | --- | --- | --- | --- | --- |
| | Model I | | Model II | | Model III | | Model IV | |
| | Coeff. | Std. Error | Coeff. | Std. Error | Coeff. | Std. Error | Coeff. | Std. Error |
| Constant | −23.82 ** | 2.68 | −12.77 ** | 1.52 | −13.87 ** | 1.45 | −7.31 ** | 0.81 |
| Separation of waste at home (1 means yes, otherwise is 0) | 0.95 ** | 0.34 | 0.92 ** | 0.32 | 0.517 ** | 0.19 | 0.45 ** | 0.17 |
| Participated in environmental protection activities (1 means yes, otherwise is 0) | 1.42 ** | 0.43 | 1.30 ** | 0.39 | 0.82 ** | 0.23 | 0.76 ** | 0.22 |
| Using eco-friendly products (1 means yes, otherwise is 0) | 0.64 * | 0.27 | 0.65 * | 0.25 | 0.36 * | 0.15 | 0.35 * | 0.14 |
| Member of environmental organizations or clubs (1 means yes, otherwise is 0) | 2.34 ** | 0.48 | 2.32 ** | 0.45 | 1.28 ** | 0.25 | 1.17 ** | 0.22 |
| Gender (1 represents female, otherwise is 0) | 0.92 ** | 0.27 | 0.85 ** | 0.25 | 0.51 ** | 0.15 | 0.45 ** | 0.14 |
| Education (1 represents education level is college/vocational or above, otherwise is 0) | 1.18 ** | 0.28 | 1.07 ** | 0.26 | 0.65 ** | 0.16 | 0.59 ** | 0.15 |
| Employment (1 represents respondent is homemaker or retired, otherwise is 0) | 1.24 * | 0.51 | 1.13 * | 0.49 | 0.72 ** | 0.27 | 0.67 ** | 0.26 |
| Residential area (1 represents urban area, otherwise is 0) | 0.85 ** | 0.30 | 1.29 ** | 0.29 | 0.49 ** | 0.17 | 0.74 ** | 0.17 |
| Overall importance mean | 4.95 ** | 0.58 | - | - | 2.90 ** | 0.32 | - | - |
| Overall performance mean | - | - | 4.11 ** | 0.55 | - | - | 2.37 ** | 0.30 |
| McFadden Pseudo $R^2$ | 0.45 | | 0.38 | | 0.45 | | 0.38 | |
| −2Log likelihood | 302.22 | | 259.84 | | 303.69 | | 256.61 | |
| AIC | 384.80 | | 427.20 | | 383.30 | | 430.40 | |
| AIC/N | 0.68 | | 0.76 | | 0.68 | | 0.76 | |
| *p*-value (Hosmer-Lemeshow test) | 0.34 | | 0.65 | | 0.44 | | 0.054 | |
| Chi square value | $\chi^2$ (0.01, 9) = 21.66 | | | | | | | |

**, *: significance at 1% and 5% levels, respectively.

Looking at the four models, it is noteworthy that citizens' perceptions of the I-P of the PWM solutions tend to exert the greatest influence on the citizens' WTP in PWM solutions, while other factors were positive but less influential. To be more specific, respondents who gave a higher rating to the importance or performance of PWM solutions were typically members of environmental organizations or clubs, participated in environmental protection activities, were homemakers or retired, had higher education levels, regularly separated waste at home, were females, lived in urban areas, and often used eco-friendly products. They were also significantly more likely than other respondents to participate in PWM solutions.

## 5. Discussion and Policy Implications

This is the first study to use the IPA method to estimate citizens' perceptions toward PWM solutions in Danang, Vietnam (Figure 4). Based on our results, there are three key observations to be made about the divergent perceptions of the respondents towards PWM solutions.

The first key finding is that urban citizens rate the importance of most PWM solutions (seven of eleven total indicators) much higher than rural citizens (Figure 3). This indicates that citizens who are living in urban areas have a higher level of concern about environmental issues and mitigation solutions, likely due to their having higher levels of education and environmental awareness (see Table 2), which is also confirmed by Bharadwaj et al. [19]

and Latinopoulos et al. [66]. In addition, environmental education campaigns in urban areas likely help citizens to become involved in PW management. Furthermore, a lack of infrastructure and the pervasiveness and extent of plastic pollution in urban areas cause the issue to be perceived as more serious than it is in rural areas, and therefore urban citizens, more than their rural counterparts, consider most PWM solutions to be important [67]. In particular, a shortage of landfill sites in Danang, as well as the pollution of the ocean with waste, are major concerns of citizens living in urban areas, as those citizens are more likely to realize that waste seriously affects the overall ecosystem, the marine environment, and human health [8].

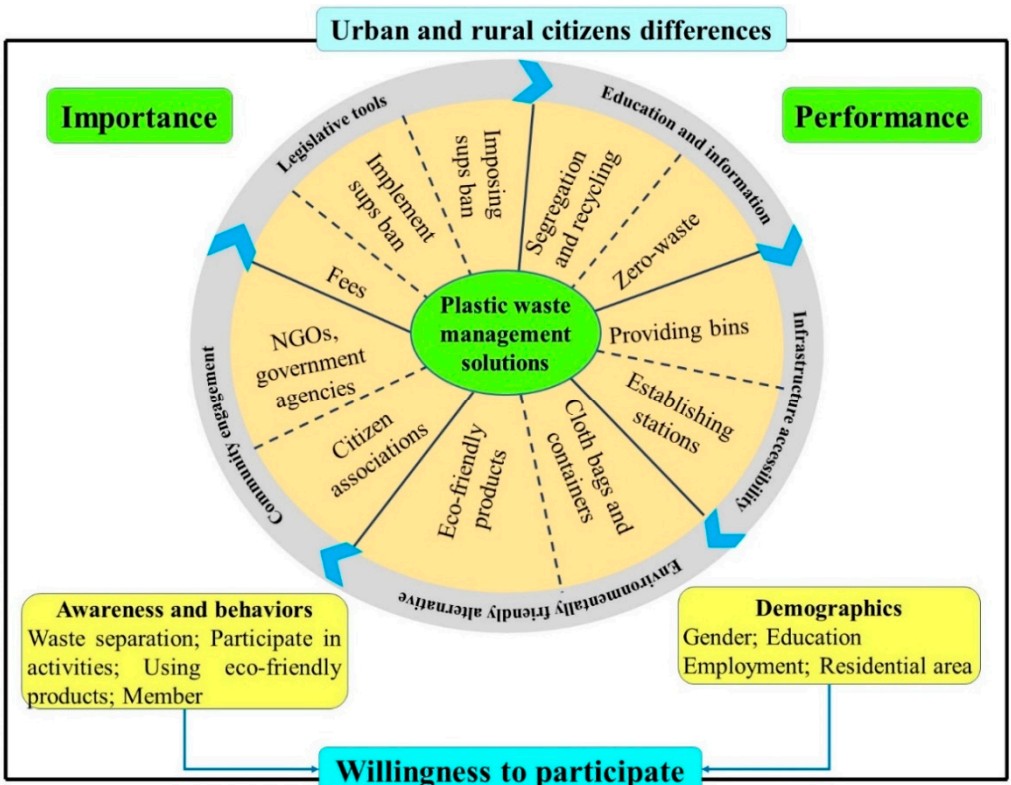

**Figure 4.** The evaluation framework of IPA for PWM.

In contrast, rural citizens argue that it is not important to implement PWM solutions, which might stem from their daily habits. Since plastic bags are convenient, functional, lightweight, and cheap, as reported in the findings of Liu et al. [13], people in rural areas prefer to use them than reusable options. In addition, those who live in remote areas find it difficult to practice PWM solutions. For example, it is inconvenient to reduce the use of plastic bags, and it takes time to separate waste [57]. Another thing worth noting is that the majority of Vietnamese people who live in rural areas have low or moderate incomes (and less economic resources, as a result) so they tend to rate PWM solutions as less important. Another reason could be the general lack of infrastructure in rural areas, which makes citizens less likely to participate in PWM programs. Overall, the differences in perception of PWM between urban and rural citizens are largely due to their awareness of PWM, education levels, economic conditions, and differences in access to infrastructure.

Noticeably, in this study, citizens gave poor performance ratings to most indicators (i.e., eight of eleven total indicators, for both urban and rural citizen cohorts) (Figure 3). This might be explained by suggesting that the effectiveness of the implementation of PWM solutions is low in this part of Vietnam. In fact, while some Vietnamese people have knowledge and awareness of the harmful effects of PW on the environment and human health, they typically have limited understanding of how to properly reduce, separate, recycle, and dispose of PW. In addition, the existing infrastructure (e.g., collection,

transportation, and treatment equipment and facilities) is not well managed, and the price associated with collection is not considered reasonable. Typically, private contractors collect waste and dump it in illegal locations [67]. Furthermore, the implementation of fees and plastic ban policies are still novelties in the Vietnamese context, reflecting the fact that many citizens might face economic difficulties if and when these policies are implemented.

The second key finding is that the majority of indicators are rated by urban citizens as quadrant II (concentrate here), while those rated by rural citizens are located in quadrant III (low priority) (Figure 3). This indicates that urban citizens perceive most PWM solutions to be highly important but performing poorly, while rural citizens consider most PWM solutions to be both less important and less well-performing. Based on these findings, it is suggested that local authorities in urban areas must focus on strengthening, prioritizing, and investing more resources in PWM solutions that are located in quadrant II, such as "zero-waste" (code I2), "NGOs, government agencies" (code I8), "establishing stations" (code I4), "eco-friendly products" (code I6), "citizen associations" (code I7), "implement SUPs ban" (code I10), and "imposing SUPs ban" (code I11). Additionally, some PWM solutions that are located in quadrant I, such as "segregation and recycling" (code I1), "providing bins" (code I3), and "cloth bags and containers" (code I5) should be continually maintained to take advantage of their strengths. Regarding rural areas, the government ought to focus on the indicator "establishing stations" and maintain efforts on the indicators "segregation and recycling" and "providing bins".

The third key finding is the significant influence, firstly, of respondents' perceptions of PW solutions and, second, of their socio-demographic characteristics, on their WTP in PWM programs. Since successful PWM frequently starts at the individual and household levels, citizens' perceptions have important implications for both the existence and persistence of the issue of PWM, and for potential solutions. A positive attitude on the part of citizens towards participation in pollution mitigation schemes will reduce the amount of PW entering the environment. Notably, our research shows that citizens' perceptions of the I-P of PWM solutions has the most significant influence of any factor on citizens' WTP in PWM strategies to prevent PW from entering the environment. Another important finding that has come to light in this study is that respondents who are females, possess higher levels of education, are members of environmental organizations, and participate in environmental protection activities are more likely than other respondents to engage in PWM solutions. This result is also supported by Chung and Yeung [59], Liu et al. [13], and Madigele et al. [42], all of whom have indicated that education is one of the key factors that leads to the formation of attitudes and perceived control. Further confirmatory evidence has been found by Ekere et al. [60], Nguyen et al. [49], and Madigele et al. [42], who have shown that females are more likely than males to accept a shift to eco-friendly alternatives to mitigate plastic entering the environment and landfills.

According to our logit and probit model findings, citizens who live in urban areas, separate waste at home, use eco-friendly products, and are homemakers or retired have a higher WTP in PWM solutions compared with other respondents. This finding is similar to results reported by Xiao et al. [57], who stated that citizens living in urban areas with more environmentally friendly behaviors have a greater inclination to participate in sustainable waste management. Arı and Yılmaz [58] concluded that homemakers are more engaged in waste separation at home. It has also been suggested that citizens' participation in PW separation and reduction is crucial for sustainable PWM and the long-term public adoption of PWM mitigation strategies [22,39,53,56]. Gill et al. [23] and Hao et al. [24] point out that citizens who regularly use eco-friendly products have a greater incentive and willingness to recycle and pay for waste management services. In light of this, it can be concluded that environmental managers and policymakers could design a platform to encourage citizens' participation based on the I-P of PWM solutions while taking into account citizens' socioeconomic characteristics (i.e., gender, residential area, education level, and employment). Additionally, managers can also understand citizens' awareness by looking at their behavior in terms of waste separation, participating in environmental

activities, and using eco-friendly products. More specifically, citizens who have a higher awareness of the impact of PW could be expected to be more likely to participate in PWM solutions. Therefore, as evidenced by the results of the goodness of fit (GOF) test, which show that, in terms of both theoretical aspects and model specificity, our findings accurately grasp the plastic waste management situation, our assessment of IPA for PWM can serve as the basis for further waste management strategies in Danang, Vietnam.

## 6. Conclusions

Our research is valuable as a starting point for the development of adequate, relevant strategies to address current and future plastic pollution, based on an actual assessment of citizens. The outcomes demonstrate a clear need for a PWM system that is appropriate for the current economic and societal environment of Vietnam. This study contributes to both a theoretical approach and a practical managerial policy on PWM by developing an evaluation framework and exploring key aspects affecting the perceived importance and performance of a waste management system. It provides three unique contributions to the development of PWM strategy. Firstly, we develop an assessment framework for PWM solutions by applying the IPA method, in which citizens are asked to evaluate various aspects. Integrating the aspects that make up these sustainable PWM solutions, we divide them into five groups: education and information, infrastructure accessibility, environmentally friendly alternatives, community engagement, and legislative tools (shown in Table 1 and Figure 4). Second, according to the IPA evaluation framework, our study examines the matrix of the I-P levels of PWM solutions to determine differences in perceptions of the eleven PWM solutions among the entire cohort, and the urban and rural citizen cohorts individually, based on demonstrably sound theoretical constructions. Finally, this study determines the factors that influence citizen's WTP in PWM strategies based on their social backgrounds, perceptions of, and behavior toward PWM solutions, as confirmed by the GOF of our model specification. The results also prove that guidelines and action plans should be provided to the citizens as a useful reference for making better decisions and assessing the sustainability of PWM operations. Additionally, the government should enhance community efforts to increase citizens' awareness of environmental protection, ocean protection, human health and green consumption choices.

**Author Contributions:** T.T.T.P. contributed to the conception and design, theoretical framework formulation, data collection, data analysis, and writing of the manuscript. C.-H.L. was responsible for the conceptualization, methodology, software, writing, and editing. V.V.N. contributed to the data analysis and edited the manuscript. H.T.T.N. managed data collection and contributed to manuscript editing. All authors have read and agreed to the published version of the manuscript.

**Funding:** This work was supported by the Ministry of Science and Technology, Taiwan [grant number 111-2621-M-259006-; 109-2628-M-259-001-MY3; 108-2410-H-259-042].

**Institutional Review Board Statement:** Not applicable. However, the research group obtained informed consent from all participants and assured them that no harm would come from participating or not in the study; participants' information is anonymous and confidential.

**Informed Consent Statement:** Informed consent was obtained from all subjects involved in the study.

**Data Availability Statement:** The illustration of PWM is from the website of MDPI.

**Acknowledgments:** The authors would like to thank the Ministry of Science and Technology, Taiwan for the funding provided for this study. The authors would like to especially thank all members of the interview team who helped us to collect data during the survey. We also thank the experts, managers, and researchers who took their time to respond to the interviews and questionnaires. We have ensured that participants' information is anonymous and confidential. We also would like to thank anonymous reviewers for their valuable comments.

**Conflicts of Interest:** The authors declare no conflict of interest.

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
