# Peer review of "Integrating Citizens’ Importance-Performance Aspects into Sustainable Plastic Waste Management in Danang, Vietnam"

_sustainability, doi:10.3390/su141610324_

Round 1
Reviewer 1 Report
Thank you for reporting this research on plastic waste management. This is a global problem that has to be solved if sustainability is to be achieved. Citizen's perception of plastic waste management is critical for success of the program and your paper is a great contribution to solving PWM worldwide.
Define what plastics you are referring to including polyethylene both low and high density, polypropylene and any biodegradable plastic component such as lactic acid.
Did any of the participants work with plastics in its production, in agriculture or in shipping business using significant amounts of plastics.
My last comment is did you ask questions regarding biodegradable plastics and their future use in Vietnam.

Author Response
Dear Reviewer,
The manuscript has been revised according to your benefitical comments and suggestions showed in the Review Report and manuscript. Many thanks for your time and efforts!
Yours sincerely,

Reviewer 2 Report
This manuscript looks well written, however, I suggest some minor revision as follows:
Line 245: t-tests with p < 0.05
(Comments: Variables should be italic, e.g., “p” and maybe “t” also)
Line 261, Figure 2. Y-axis’s title should be rotate 180 degree.
(Indicator importance => Rotate it)
Line 283. VDM => Full name should be introduced in this line.
Line 359. Figure 3. Authors should input both x-axis and y-axis’ titles for reader-friendly
Line 375. Chi-square => Here, Authors should introduce “χ” symbol, e.g., χ2
Line 488. GOF test => Full name should be introduced here.
Technical comment>>
Authors mentioned “biodegradable products”. If these biodegradable products are plastics (polymers), (1) please mention some examples and (2) describe which biodegradable plastic can replace which non-degradable polymer.
Line 409-410, “Since plastic bag is convenient, functional, lightweight, and cheap, as reported in the findings of Liu, et al”….. Question: Biodegradable plastic can meet the aforementioned criteria? If yes, please mention some examples.
Author Response

(The authors gave the same response as above.)

Reviewer 3 Report
This research claimed that plastic pollution is a matter of deep concern that requires an urgent and international response, involving stakeholders at different levels. Especially in the context of COVID-19, the rapid increase of single-use plastic and medical waste has sent the plastic pollution crisis ratcheting upward on a global scale. Based on analyses at Danang city, Vietnam, the authors have shown that solutions such as “offering zero-waste lifestyle seminars to citizens”, “having community engagement”, “using eco-friendly products”, and “imposing a ban on single-use plastics” are useful for the development of an effective environmental policy. The results have also shown that characteristics have a significant influence on citizens’ participation in PWM solutions: gender, education level, residential area, employment status, and citizens’ awareness and behaviour towards plastic reduction.
1) Table 4. Estimation results of citizens’ perception of PWM participation behaviour should be presented by more scientists.
2) A table of acronyms should be added in Section 1 for the convenience of the reader.
3) Some typos need to be checked throughout the manuscript.
Author Response

(The authors gave the same response as above.)
